# K143R Amino Acid Substitution in 14-α-Demethylase (Erg11p) Changes Plasma Membrane and Cell Wall Structure of *Candida albicans*

**DOI:** 10.3390/ijms23031631

**Published:** 2022-01-31

**Authors:** Daria Derkacz, Przemysław Bernat, Anna Krasowska

**Affiliations:** 1Department of Biotransformation, Faculty of Biotechnology, University of Wroclaw, 50-383 Wroclaw, Poland; daria.derkacz@uwr.edu.pl; 2Department of Industrial Microbiology and Biotechnology, Faculty of Biology and Environmental Protection, University of Łódź, 90-237 Lodz, Poland; przemyslaw.bernat@biol.uni.lodz.pl

**Keywords:** *Candida albicans*, ergosterol, plasma membrane, cell wall

## Abstract

The opportunistic pathogen *Candida albicans* is responsible for life-threating infections in immunocompromised individuals. Azoles and polyenes are two of the most commonly used antifungals and target the ergosterol biosynthesis pathway or ergosterol itself. A limited number of clinically employed antifungals correspond to the development of resistance mechanisms. One resistance mechanism observed in clinical isolates of azole-resistant *C. albicans* is the introduction of point mutations in the *ERG11* gene, which encodes a key enzyme (lanosterol 14-α-demethylase) on the ergosterol biosynthesis pathway. Here, we demonstrate that a point mutation K143R in *ERG11* (*C. albicans* *ERG11*^K143R/K143R^) contributes not only to azole resistance, but causes increased gene expression. Overexpression of *ERG11* results in increased ergosterol content and a significant reduction in plasma membrane fluidity. Simultaneously, the same point mutation caused cell wall remodeling. This could be facilitated by the unmasking of chitin and β-glucan on the fungal cell surface, which can lead to recognition of the highly immunogenic β-glucan, triggering a stronger immunological reaction. For the first time, we report that a frequently occurring azole-resistance strategy makes *C. albicans* less susceptible to azole treatment while, at the same time, affects its cell wall architecture, potentially leading to exposure of the pathogen to a more effective host immune response.

## 1. Introduction

*Candida albicans* (*C. albicans*) constitutes a natural part of the human microbiome. However, in cases of immunodeficiency, it can cause an opportunistic infection called candidiasis [1]. According to a Centers for Disease Control and Prevention (CDC) report, there are approximately 25,000 cases of candidemia (blood infection caused by *Candida* sp.) each year, and *C. albicans* remains the dominant species isolated from the patient population [2,3]. Large numbers of *Candida*-related infections, along with ineffective treatment, lead to a high mortality rate among patients suffering from candidemia (38–75%) [4]. The major problem during anticandidal therapy is a worldwide resistance of *Candida* sp. to commonly used antifungals [5]. This is due to widely used antifungals, e.g., azoles, clinically implemented in the 1980s [6]. Decades of azole usage has led to the development of effective defense mechanisms to avoid the toxic activity of antifungals.

Azoles (e.g., fluconazole, ketoconazole) inhibit the activity of lanosterol 14-α-demethylase (Erg11p, encoded by the *ERG11* gene), a key enzyme of the ergosterol biosynthesis pathway [7]. Ergosterol constitutes a component of the fungal plasma membrane (PM) which is important for its functionality and the maintenance of its structural properties [8]. A diminished amount of ergosterol and the accumulation of toxic 14-α-methylated sterols in fungal PM after azole therapy contributes to pathogen growth inhibition [9]. The fact that azole drugs are not fungicidal but fungistatic is one of the disadvantages of azole usage and contributes to the creation of vast numbers of resistance mechanisms. An example of the resistance mechanism is the presence of point mutations in *ERG11* (especially near the active site of the enzyme), resulting in a decreased binding affinity of the azoles to Erg11p [7,10]. Another azole resistance strategy is an altered sterol composition in the PM of *C. albicans* and increased expression of *ERG11* due to the upregulation of its transcription factor *UPC2*, and the overexpression of the genes encoding multidrug resistance transporters (e.g., *CDR1*, *CDR2*) [10,11].

In contrast to azoles, polyenes (e.g., amphotericin B (AMB), nystatin) are fungicidal and bind directly to the ergosterol present in *C. albicans* PM [5]. Polyenes induce the formation of pores in the PM, leading to ion leakage and cell death [12]. Due to the characteristic mode of action of the polyenes, the resistance of clinical isolates remains a very rare event, but it can be achieved by depriving the pathogen PMs of ergosterol [13]. Other drugs that are employed in antifungal therapy are echinocandins (e.g., caspofungin) that act on the *C. albicans* cell wall (CW), affecting glucan synthesis [14]. Furthermore, caspofungin causes significant β-glucan exposure on the *C. albicans* cell surface, resulting in an increased phagocytic response by the dendritic cells [15]. The *C. albicans* CW consist of three components: chitin, β-glucan, and a thick layer of mannoproteins [16]. Typically, chitin and β-glucan are located within the inner CW, providing shape and strength [17]. Meanwhile, the outer layer of the CW is composed of mannans and cell wall proteins (CWPs) that are glycosylphosphatidylinositol (GPI)-anchored in the inner chitin/β-glucan core [18]. Recently, we have reported that the deprivation of ergosterol (as a consequence of *ERG11* gene deletion) from the *C. albicans* PM corresponds to β-glucan unmasking on the CW surface [19]. This suggests that the presence of ergosterol in fungal PM is crucial for maintaining the CW composition, and changes in the CW structure can lead to diverse host immune responses to the pathogen.

Considering the importance of ergosterol as a drug target, we investigated the influence of a nonsynonymous point mutation (*K143R/K143R*) in the *ERG11* gene on the PM and CW of *C. albicans*. This mutation was chosen since it was discovered in fluconazole-resistant *Candida* sp. clinical isolates [20,21]. Here, we report that the introduction of the point mutation in *ERG11* correlates with the increased expression of this gene. The overexpression of the *ERG11* gene results in a significantly higher ergosterol content in the PM, which increases the resistance of the fungus to commonly used antifungal drugs (e.g., fluconazole (FLC)). We also compare the sterol profile of *C. albicans ERG11^K143R/K143R^* with the *C. albicans erg11Δ/Δ* strain that was previously investigated by our team [22]. We demonstrate that both the deprivation and the elevated content of ergosterol contribute to the increased fluidity of *C. albicans* PM. Furthermore, our recent findings show that a lack of ergosterol in *C. albicans* PM caused chitin and β-glucan exposure [19], and so it is relevant to explore the influence of the K143R substitution on the *C. albicans* cell morphology. Our investigations revealed that the point mutation in *ERG11*, as well as the gene deletion, result in chitin and β-glucan unmasking on the cell surface, highlighting the crucial role of ergosterol for maintaining the PM integrity and CW architecture. Our data show that alterations in the ergosterol content in *C. albicans* PM can also correspond to significant changes in the immune response profile following antifungal therapy during a pathogen infection. β-glucan is a highly immunogenic pathogen-associated molecular pattern (PAMP) recognized by pattern recognition receptors (PRRs) (e.g., dectin-1) present on the host cell surface [23]. Unmasking of PAMPs on the fungal cell surface can result in the changed host immune response by triggering the production of a different cytokine profile [24].

## 2. Results

### 2.1. Point Mutation in ERG11 Does Not Affect the Growth of the C. albicans ERG11^K143R/K143R^ Strain and Contributes to Increased Fluconazole Resistance

In order to investigate the impact of an introduced point mutation in *ERG11* gene (*K143R/K143R*), we performed the growth curve and the azole susceptibility test for the *C. albicans ERG11^K143R/K143R^* (10C1B1I1) strain. The results were compared to those obtained for the *C. albicans* strain which lacks the *ERG11* gene (KS058).

We observed that a point mutation (*K143R/K143R*) in *ERG11* of *C. albicans* 10C1B1I1 did not affect the growth of this strain, and no temporal shift in growth phases was observed (Figure 1A). As opposed to *C. albicans* 10C1B1I1, the KS058 strain (*erg11Δ/Δ*) exhibited a significantly lower growth rate compared to the *C. albicans* WT strain. This impaired growth of *C. albicans erg11Δ/Δ* was previously reported by our team, but for the parental CAF2-1 strain [22]. Interestingly, the single amino acid replacement in Erg11p contributes to the decreased susceptibility of *C. albicans* 10C1B1I1 to FLC (IC50 eight times higher than for WT), but not to amphotericin B (AMB) (Figure 1B), and this is likely due to the less-effective binding of azole to the active site of the enzyme. Considering that *C. albicans* KS058 lacks targets for both FLC (Erg11p) and AMB (ergosterol), the *ERG11* gene deletion correlates with resistance toward the investigated antifungals.

### 2.2. The Point Mutation ERG11^K143R/K143R^ Results in an Increased ERG11 Gene Expression in the C. albicans 10C1B1I1 Strain

Taking into account that introducing the *K143R/K143R* point mutation in *ERG11* resulted in a decreased sensitivity toward FLC, we decided to investigate whether it will also affect the *ERG11* gene expression at different phases of growth (8, 14 and 24 h) and after FLC or AMB treatment (Figure 2). The concentrations of FLC (4 μg/mL) and AMB (0.25 μg/mL) were selected based on the viability test (Figure 1B), and these are IC50 (for FLC), or two times lower than the IC50 (for AMB) for the 10C1B1I1 strain.

Our analysis revealed that the *C. albicans* 10C1B1I1 strain exhibits an increased *ERG11* gene expression under control conditions (YPD medium alone) in the early (8 h) and late (14 h) logarithmic growth phases. After treatment with FLC, the *ERG11* expression was slightly reduced compared to the control conditions at 8 h of growth for the *C. albicans* 10C1B1I1 strain. However, at the stationary growth phase (24 h), the *ERG11* expression in 10C1B1I1 was two-fold higher than in the WT strain. Interestingly, the treatment of the 10C1B1I1 strain with AMB resulted in a significantly higher *ERG11* gene expression in the stationary growth phase (three-fold higher than WT strain). The expression of *ERG11* is also significantly higher following 24 h of culture after the AMB treatment than in the control conditions (Figure 2).

### 2.3. The Overexpression of the ERG11 Gene Corresponds to an Elevated Ergosterol Content in the C. albicans ERG11^K143R/K143R^ Strain

We decided to further investigate whether the overexpression of *ERG11* in the 10C1B1I1 strain contributes to an increased level of ergosterol. In order to verify whether the *K143R/K143R* point mutation in the *ERG11* gene resulted in an altered sterol profile of *C. albicans*, we performed GC-MS analysis. We performed the same analysis for *C. albicans erg11Δ/Δ* in order to compare our results with those obtained for *C. albicans ERG11^K143R/K143R^* (Table 1).

At the early logarithmic (8 h) growth phase, the level of ergosterol in 10C1B1I1 was comparable to the WT strain. In both cases, level of ergosterol significantly increased with culture aging and the ergosterol content in PM of the *C. albicans* 10C1B1I1 strain at 14 h was 1.9-fold higher than the WT strain (93.82 and 49.49 μg/mg, respectively). The level of ergosterol decreased in both strains during the stationary phase growth (24 h), but for the 10C1B1I1 strain, the levels remained 1.3-fold higher than for WT strain (37.18 and 28.82 μg/mg, respectively) (Table 1).

The FLC treatment resulted in an absence of ergosterol in the PM of the *C. albicans* WT strain in all growth phases, although a trace amount of ergosterol was found in the PM of the WT cells in the stationary growth phase (0.73 μg/mg). For the *C. albicans* 10C1B1I1 cells, the level of ergosterol found in an 8 h culture after FLC treatment was comparable to those found in the control conditions (5.83 and 7.84 μg/mg, respectively). A significant decrease in the ergosterol level was observed in both 14 h and 24 h of 10C1B1I1 growth (82.2 and 85% decrease in ergosterol, respectively; *p* < 0.001) compared to control conditions.

The addition of AMB to both the WT and 10C1BI1 cultures resulted in a significant increase in ergosterol in the 8 h growth phase compared to the control conditions (around 5-fold change in the ergosterol content for both *C. albicans* strains). For *C. albicans* 10C1B1I1, this may be the consequence of the overexpression of *ERG11* (Figure 2). The ergosterol level in the 10C1B1I1 strain treated with AMB decreased with the aging of the culture and was comparable to the WT strain.

Our results also demonstrate a significantly higher lanosterol content for the *C. albicans* KS058 strain in control conditions (four-fold higher at 8 h of culture compared to the WT and 10C1B1I1 strains). At 14 h of culture, a lower lanosterol content was noted for the WT and 10C1B1I1 strains (2.52 and 4.04 μg/mg, respectively) and its level was comparable to that obtained for the 24 h cultures. At both culture times (14 h and 24 h of culture), lanosterol concentration was significantly higher for the KS058 strain (11.45 and 22.53 μg/mg, respectively; *p* < 0.001) compared to the WT and 10C1B1I1 strains.

During our investigations, eburicol was detected only after the AMB treatment of 10C1B1I1 at 8 h and 14 h of culture and after the FLC treatment for both the WT and 10C1B1I1 strains. However, elevated levels of eburicol were observed for the KS058 strain when compared to the WT and 10C1B1I1 strains in all investigated conditions. The eburicol content at 8 h and 14 h in the KS058 strain culture exhibited similar levels (9.41 and 9.47 μg/mg, respectively) and increased at 24 h of culture (40.07 μg/mg).

Levels of 14-α methylergosta-8-24(28)dienol were detected in all culture conditions for the *C. albicans* KS058 strain, while for the WT and 10C1B1I1 strains, it was only detected at 24 h of culture after the FLC treatment. Similar to lanosterol, an increased level of 14-α methylergosta-8-24(28)dienol was also observed previously by our team in *C. albicans* KS028 [22]. However, the level of 14-α methylergosta-8-24(28)dienol detected in the KS058 strain (parental to the SC5314 strain) was lower compared to the KS028 strain (parental to CAF2-1 strain).

### 2.4. An Increased Ergosterol Content in C. albicans ERG11^K143R/K143R^ Results in Decreased Plasma Membrane Fluidity

To investigate whether an increased ergosterol content in the *C. albicans* 10C1B1I1 strain leads to altered PM fluidity, we measured the fluorescence of a Laurdan probe incorporated to the PM of the *C. albicans* cells of all strains analyzed in this work (Table 2).

Our analysis showed that the *C. albicans* KS058 and 10C1B1I1 strains at 8 h of culture exhibit significantly higher GP values (−0.21 and −0.10, respectively), highlighting the correlation between both the deprivation and the overproduction of ergosterol and decreased PM fluidity. The representative, normalized fluorescence intensities spectra for the investigated strains in 8 h or 24 h of culture are presented in Figure 3.

For *C. albicans* KS058 and 10C1B1I1, we detected a red shift for Laurdan IF at 8 h of culture (Figure 3A). The PM fluidity of the investigated strains decreased significantly at 24 h of culture (Table 2) and this was expressed in a deepened red shift of the Laurdan IFs (Figure 3B). However, the overproduction of ergosterol in the 10C1B1I1 strain resulted in 1.4-fold lower fluidity of the PM (positive GP value) comparing to the WT and KS058 strains.

### 2.5. The Deletion and Increased Expression of the C. albicans ERG11 Gene Correlates with CW Remodeling

In order to verify whether the increased production of ergosterol in the 10C1B1I1 strain of *C. albicans* contributes to CW remodeling, we performed staining for all three components (chitin, β-glucan, and mannan) (Figure 4).

Microscopic study revealed that chitin is typically located in the inner part of the CW of the *C. albicans* WT strain, covered by a thick layer of mannans (Figure 4A). A different situation was observed for the KS058 strain, where the chitin layers overlap with the mannans. At the bud scars, there was a pointwise fluorescence signal from the exposed β-glucan. This demonstrate that chitin and β-glucan could be partially unmasked in the KS058 strain, and unmasked chitin was observed in the region of the bud scar of the 10C1B1I1 strain. Similar to the KS058 strain, the exposed β-glucan accumulation was detected within the bud scar of the 10C1B1I1, but the fluorescence signal was also present along the full fungal cell length. In addition, we determined the exposure rate of the chitin and β-glucan at the bud scars for *C. albicans* WT, KS058, and 10C1B1I1 (Figure 4B).

Analysis revealed that changes in the PM caused by the altered ergosterol level correlates with the remodeling of *C. albicans* CW. β-glucan, typically hidden under the mannoprotein layer (*C. albicans* WT), was unmasked in both the *C. albicans* KS058 and 10C1B1I1 strains at the region of the bud scar (Figure 4B). In the case of the *C. albicans* WT strain, chitin was located in the inner part of the CW. Opposite to the WT, for both *C. albicans* strains with the depletion and overproduction of ergosterol, we observed a shift of the chitin layer to the outer part of the CW, and it partially overlapped with the mannans and the β-glucan.

We also performed a FACS analysis to verify the amount of particular CW components (unmasked chitin, β-glucan, and mannans) present after 24 h of culture (Figure 5).

Analysis revealed the significant exposure of chitin and β-glucan in both the KS058 and 10C1B1I1 strains. This is supported by the detection of β-glucan signals in the most exposed fragment of KS058 and 10C1B1I1 CW in contrast to the WT (Figure 5). In the case of unmasked chitin (Figure 5A), we observed an MFI value for KS058 and 10C1B1I1 that is 3-fold and 1.3-fold higher, respectively, than for the WT strain, while exposed β-glucan MFIs are 17-fold and 1.3-fold higher for the KS058 and 10C1B1I1 strains, respectively (Figure 5B). This may be related to the local accumulation of unmasked β-glucan in the bud scars region of KS058. Our research indicates that the 10C1B1I1 strain possess a decreased amount of mannans (MFI 1.4-fold lower than the WT and KS058 strains; Figure 5C). Herein, we also demonstrated that increased ergosterol contributes to chitin and β-glucan exposure on the outside of the fungal cell, as well to a decreased mannan content (Figure 4 and Figure 5).

## 3. Discussion

One of the most common azole-resistance strategies of *C. albicans* is introducing a point mutation in the gene (*ERG11*) encoding enzyme Erg11p—the target for azole drugs [7]. The K143R amino-acid substitution in Erg11p is considered to be responsible for alterations to the tertiary structure of Erg11p, resulting in a lower binding affinity of azoles to this enzyme [25]. This is due to the fact that lysine K143 is located in the exposed active-site cavity of the enzyme, and amino-acid replacement in this hot-spot region could correspond to conformational change in the protein [26]. Our research revealed that this single amino-acid substitution caused increased resistance toward FLC, but not to AMB (Figure 1B), indicating the role of the *K143R/K143R* point mutation in resistance toward first-choice antifungals such as azoles. Furthermore, the lysine 143 substitutions (e.g., K143R and K143Q) were also identified in the azole-resistant *Candida* spp. clinical isolates [20,24], demonstrating the importance of this amino-acid for the proper binding of azoles to Erg11p.

Our analysis also revealed a higher expression of the *ERG11* gene particularly in the 8 h and 14 h of culture (Figure 2). The increased *ERG11* gene expression and the reduced azole susceptibility were associated with the upregulation of the transcription factor (TF) of this gene (e.g., *UPC2*) [27]. In the case of K143R substitution, it is highly unlikely that TF has an influence on the increased expression of *ERG11*, as *UPC2* binds the *ERG11* promoter sequence [28]. Nevertheless, synonymous codon substitutions in *ERG11* were shown to lead to the increased expression of the *ERG11* gene in *C. krusei* [29]. Thus, a similar situation can occur in the case of K143R substitutions in Erg11p. The increased expression of the *ERG11* gene after FLC or AMB treatment (Figure 2) could be the result of a stress response due to used antifungals. Here, we report for the first time that K143R substitution in Erg11p not only confer FLC resistance, but also cause an increase in the expression of the *ERG11* gene, leading to higher ergosterol production during FLC treatment.

Furthermore, sterol analysis revealed the significant increase in the ergosterol content in the late logarithmic and stationary phases of growth in *C. albicans ERG11^K143/K143R^* (Table 1). Taking this into account, the increased *ERG11* gene expression in the early and late logarithmic growth phases correlates with an elevated ergosterol content.

Current work demonstrates that point mutation in *ERG11* results not only in a lower susceptibility to FLC, but also contributes to higher levels of ergosterol after FLC treatment (Table 1). The addition of FLC (4 μg/mL) results in the absence of ergosterol in *C. albicans* WT, but not in *C. albicans ERG11^K143/K143R^*. This could be the consequence of an increased expression of the *ERG11* gene and the overall elevated level of Erg11p. Furthermore, the inhibition rate for the mutated form of an enzyme is not as effective as for the WT form of the protein [30]. Therefore, the K143R substitution not only contributes to higher *ERG11* expression (Figure 2), but also provides FLC resistance because of an elevated level of ergosterol.

The data demonstrates that increased levels of ergosterol do not contribute to AMB resistance. Cellular stress caused by AMB results in the elevated production of ergosterol in the early growth phases of AMB treatment. Conversely, supplementation of the *C. albicans* culture with ergosterol in the presence of AMB had been shown to inhibit the release of K^+^ ions from fungal cells [31]. Our analysis revealed that the K143R substitution contributes to increased FLC resistance (Figure 1B); however, in cultures treated with AMB, this mutation leads to the overproduction of ergosterol, resulting in the enhanced binding of AMB to the *C. albicans* cell and the sensitization of the fungal cells to polyenes [32].

After 24 h culture of *C. albicans erg11Δ/Δ*, about a 10-fold increase of lanosterol was detected compared to the WT strain (Table 1). This suggests that the lack of ergosterol in the *erg11Δ/Δ* strain is compensated by the presence of increased lanosterol in the PMs. An elevated lanosterol content was previously described by our team, but for the *C. albicans* KS028 strain (*erg11Δ/Δ*, isogenic to the CAF2-1 parental strain) [22]. KS058 lacks the targets for FLC and AMB (Erg11p and ergosterol, respectively), so treatment of the KS058 strain with FLC and AMB would not contribute to a significant change in the lanosterol content in any of the culture conditions except for the 24 h of culture after FLC treatment. In this condition, a two-fold increase in the lanosterol content was observed compared to control conditions. Furthermore, depriving *C. albicans* PM of ergosterol results in the presence of both eburicol and 14-α methylergosta-8-24(28)dienol (Table 1). For *C. albicans erg11Δ/Δ*, the consequence of the interruption of ergosterol production is the accumulation of lanosterol. This sterol is converted by Δ(24)-sterol C-methyltransferase (Erg6p) into eburicol, then by D5,6-desaturase (Erg3p) into 14-α methylergosta-8-24(28)dienol [33]. This explains the presence of eburicol, 14-methylergosta-8,24-diendiol, and an elevated concentration of lanosterol in the KS058 strain. Low concentrations of eburicol were found in the WT and 10C1B1I1 strains only after FLC treatment, or after 8 h and 14 h of cultures with AMB for the 10C1B1I1 strain. 14-α methylergosta-8-24(28)dienol was present only after a 24 h culture with FLC for the WT and 10C1B1I1 strains, and its level was around six-fold lower than for *C. albicans* KS058.

Considering the alterations in the PM sterol profile shown in this work, we decided to determine the impact of these changes on the PM properties. Our analysis of the PM fluidity of *C. albicans* indicated significantly higher GP values for the *C. albicans* KS058 and 10C1B1I1 strains at 8 h culture (Table 2) compared to the WT strain, expressed as a red shift for Laurdan IF (Figure 3A). This effect was deepened at 24 h of culture, resulting in a red shift of Laurdan IFs (Figure 3B). An elevated level of ergosterol in the 10C1B1I1 strain resulted in a positive GP value, underlining the crucial role of ergosterol for PM properties.

The increased level of sterols such as lanosterol or eburicol corresponds to the decreased PM fluidity of KS058 when compared to the WT strain. FLC and AMB treatment also results in decreased GP values for all strains at 8 h and 24 h of culture, compared to control conditions. Interestingly, these antifungals also cause the increased fluidity of the KS058 strain PM, despite the absence of targets for these therapeutics (Erg11p and ergosterol, respectively). This could be a consequence of the increased lanosterol and eburicol content in the PM of the KS058 strain (Table 1). The rigidity of the PM in artificial bilayers can be affected by the structural motifs of sterols, and it generally increases with the elevated sterol content [8]. It has been proven that the treatment of *S. cerevisiae* with FLC corresponded to the reduction of PM rigidity, which led to a decreased order of lipids in the PM [34]. This suggests that the altered sterol composition of fungal PM after azole treatment impacts the lipid packaging, resulting in the decreased fluidity of the PM.

Our recent findings indicate deprivation of ergosterol from *C. albicans*’ PM results in both chitin and β-glucan unmasking and the elevated expression of the genes involved in chitin (e.g., *CHS3*, *CHS4*) and β-glucan synthesis (e.g., *GSC1*, *KRE9*) [19]. Nevertheless, similar remodeling of the CW composition was never described for *C. albicans* strains carrying the single amino-acid substitution. In order to investigate whether the K143R mutation in Erg11p correlates with the altered CW structure, we applied quantitative analysis to all components of *C. albicans* CW.

Microscopic study confirmed the alterations of the *C. albicans* KS058 and 10C1B1I1 CW compositions (Figure 4). After cytokinesis, the chitin layer is rebuilt in bud scars by specific enzymes [35], and so we observed a typical accumulation of chitin in this region of the cell. In the case of *C. albicans* KS058 (*erg11Δ/Δ*), chitin is located near the mannans layer, in contrast to the *C. albicans* WT strain, where the chitin and mannans layers are separated (Figure 4). For *C. albicans* 10C1B1I1, chitin partially overlaps with the mannans signal (area of the bud scar). Furthermore, this study confirmed that in the *C. albicans* WT strain, exposed β-glucan is barely visible, indicating that it is normally hidden in the inner layers of the CW [16]. For both the KS058 and 10C1B1I1 strains, the unmasked β-glucan was present in the bud scar region. In contrast to the KS058 strain, for the 10C1B1I1 strain, exposed β-glucan was also present along the full fungal cell length. FACS analysis confirmed the partial exposure of chitin and β-glucan for the *C. albicans* ergosterol mutants (Figure 5). Additionally, a lower mannan content was detected for the *C. albicans* 10C1B1I1 strain. This can be the consequence of the chitin and β-glucan layers shifting to the outer part of *C. albicans* CW (Figure 4).

So far, *C. albicans* CW remodeling has been described as a result of varying culture conditions (e.g., the addition of lactate to the culture medium) or a stress response to an antifungal drug treatment (e.g., caspofungin therapy) [14,36]. Here, we demonstrate that both the deprivation and the increased amount of ergosterol leads to the altered *C. albicans* CW composition. Furthermore, K143R substitution not only resulted in changes in the PM properties (decreased fluidity, elevated ergosterol content), but also in the CW composition. This demonstrates that one of the most common azole-resistance strategies (point mutation in *ERG11* gene) has a pleiotropic effect on the cell morphology of *C. albicans*. It should be noted that rearrangement of *C. albicans* CW could be crucial for the recognition of the pathogen by host PRRs. Thus, it is relevant to further investigate the impact of changes to the immune response that arose in *C. albicans* KS058 and 10C1B1I1 ergosterol mutants.

## 4. Materials and Methods

### 4.1. Reagents and Chemicals

The reagents and chemicals used in this study were obtained from the following sources: fluconazole, Laurdan, β-mercaptoethanol (BME), albumin fraction V (BSA), ethylenediaminetetraacetic acid (EDTA), phosphate buffered saline (PBS) tablets, CFW, WGA-FITC, ConA-FITC-conjugated, formaldehyde (Merck Life Science; Poznań, Poland); ConA (Vector Laboratories, Burlingame, USA); DNase I (Fermentas, distributor: Thermo Fisher, Waltham, MA, USA); High-Capacity cDNA Reverse Transcription Kit (manufacturer: Applied Biosystems, distributor: Thermo Fisher, Waltham, MA, USA); Total RNA Mini Kit (A&A Biotechnology, Gdańsk, Poland); Alexa Fluor 568-conjugated streptavidin (Invitrogen, distributor: Thermo Fisher, Waltham, MA, USA); iTaq Universal SYBR Green Supermix (Bio-Rad, Hercules, CA, USA); conventional AMB, D-glucose, bacteriological agar, zymolyase, D-sorbitol, (manufacturer: BioShop; distributor: Epro Science, Puck, Poland); peptone, yeast extract (YE) (manufacturer: BD; distributor: Life Technologies; Warszawa, Poland); Fc–hDectin-1 (Invivogen, San Diego, CA, USA); Alexa Fluor 448-conjugated anti-human IgG Fc antibodies (Thermo Fisher, Waltham, MA, USA); N,O-Bis(trimethylsilyl)trifluoroacetamide with trimethylchlorosilane (Merck, Darmstadt, Gemany); water (Merck); hydrochloric acid (HCl), potassium hydroxide (KOH), hexane (Merck); chloroform (CHCl3), methanol (MetOH) (Chempur; Piekary Śląskie, Poland).

### 4.2. Strains and Culture Conditions

Strains used in this study were: *C. albicans* SC5314, which was kind gift from Prof. D. Sanglard (Lausanne, Switzerland) [37], and *C. albicans* 10C1B1I1, which was kind gift from Prof. D. Rogers (Department of Clinical Pharmacy, University of Tennessee Health Science Center, Memphis, TN, USA) (genotype: *ERG11K143R::FRT/ERG11K143R::FRT*, previously named *C. albicans* 10C1B1M1) [38]. *C. albicans* KS058 genotype is the same as SC5314, but called *erg11Δ::SAT1-FLIP/erg11Δ::FRT* [22]. Strains were maintained in a YPD (yeast extract-peptone-dextrose) medium containing 1% yeast extract, 1% peptone, and 2% glucose. Two percent agar was used to solidify the medium. Strains were pre-cultured in the YPD medium, shaking (120 rpm) at 28 °C, 24 h. For specific experiments, cells were grown in 20 mL YPD (starting A_600_ = 0.1; 28 °C; 120 rpm; with or without the addition of FLC or AMB) for 8, 14 or 24 h, depending on the experiment.

### 4.3. The Determination of Growth Phases (Growth Curve)

In order to determine the growth phases, the *C. albicans* strains were pre-grown in the YPD medium (28 °C; with shaking: 120 rpm) overnight. Then 100 μL of fresh YPD medium was added to the sterile 96-well plate (Sarstedt) and inoculated with *C. albicans* pre-culture to the final A_600_ = 0.1. The A_600_ was registered for 24 h at 1 h intervals using the Spark multimode microplate reader (Tecan, Männedorf, Switzerland). The experiment was independently replicated three times.

### 4.4. Minimal Inhibitory Concentration (MIC)

For the purpose of the determination of the MIC values, a serial dilution of FLC or AMB was carried out at concentrations ranging from 0 to 256 or 16 μg/mL, respectively. The YPD medium, supplemented with different concentrations of FLC or AMB, was inoculated with *C. albicans* overnight culture at a final A_600_ = 0.01 using sterile 96-well plates. The plates were then incubated for 24 h at 28 °C and the optical density was measured using a plate reader at λ= 600 nm (Asys UVM 340, Biogenet, Cosenza, Italy). The negative and growth control groups were indicated by wells containing the YPD medium without tested compounds.

### 4.5. Real-Time Quantitative PCR (RT-qPCR) Reaction

*C. albicans* pellets were collected (A_600_ = 20) and RNA isolated using a Total RNA Mini Kit according to the manufacturer’s instructions. The concentration and purity of the isolated RNA was measured using a NanoDrop 2000 Spectrophotometer (Thermo Fisher Scientific, Waltham, MA, USA). In order to remove genomic DNA, samples were treated with DNAse I. The isolated RNA from all samples was brought to an equal concentration (14 ng/μL) and the cDNA was synthesized using a High-Capacity cDNA Reverse Transcription Kit.

The reaction was carried out using specific primers for genes *ACT1* (reference gene) and *ERG11* as follows: ACT1F (5′-TCCAGCTTTCTACGTTTCCA-3′), ACT1R (5′-GTC AAGTCTCTACCAGCCAA-3′), ERG11F (5′-TTTGGTGGTGGTAGACATA-3′), ERG11R (5′-GAACTATAATCAGGGTCAGG-3′). RT-qPCR reaction was conducted using the iTaq Universal Sybr Green Supermix and the StepOnePlus Real-Time PCR System (Applied Biosystems, Waltham, MA, USA). The initial step of the thermal cycling program was performed at 95 °C for 10 min, followed by 40 cycles at 95 °C for 20 s, 45 °C for 20 s, and 72 °C for 30 s. The determination of gene expression levels was performed as described previously using the 2^–∆∆Ct^ method [39].

### 4.6. Plasma Membrane Isolation and Sterol Analyses

The PM samples were isolated according to the previously reported method, with some modifications [40]. Briefly, cell sediment from *C. albicans* SC5314, 10C1B1I1, and KS058 after 8, 14 and 24 h cultures (A_600_ = 40) were resuspended in a lysis solution (1 M sorbitol, 0.1 M EDTA, 1% BME, 3 mg/mL zymolyase) and incubated for 30 min at 37 °C. Protoplasts were washed with 1.2 M sorbitol, then ice-cold H_2_O_dd_ was added and lysed using sonication (5-s cycles, 2 min each; 4 °C) using an ultrasonic processor (Heilscher UP50H). The lysed cells were centrifuged (10 k rpm; 10 min; 4 °C) in order to remove the unbroken lysate. The supernatant was ultracentrifuged (100 k rpm; 60 min; 4 °C) using a Micro Ultracentrifuge CS150FNX (Hitachi; Tokyo, Japan). The crude PM pellets were resuspended in phosphate-buffered saline (PBS) with CHCl_3_–MetOH (1:2 *v*/*v*). The CHCl_3_ phase was transferred to glass vials and concentrated using nitrogen gas after continuous stirring at 4 °C for 48 h.

### 4.7. GC-MS Sterol Analysis in the Plasma Membrane

The work-up procedure was previously described [22]. We added to the concentrated lipid extracts 0.5 mL CHCl_3_, 0.5 mL MetOH-KOH (0.6 M), and 20 μL cholesterol solution in CHCl_3_ (calibration standard, 1 mg/mL). The samples were vortexed and incubated at 23 °C for 1 h. Then, 0.325 mL 1M HCl and 0.125 mL H_2_O were added and centrifuged (5000 rcf; 10 °C; 5 min). The lower chloroform layer containing the lipids was transferred to 1.5 mL Eppendorf tubes and dried. Then, 100 μL of silylation reagent BSTFA + TMCS was added and heated for complete silylation at 85 °C for 90 min. The cooled samples were supplemented with 50 µL of hexane and vortexed. An analysis was performed using a gas chromatograph (Agilent 7890) equipped with HP-5ms columns (30 m × 0.25 mm inner diameter, i.d. ×0.25 mm film thickness, f.t.) and a 5975C Mass Detector. The column was maintained at 100 °C for 0.5 min^−1^, then increased to 240 °C at a rate of 25 °C min^−1^, and finally to 300 °C at a rate of 3 °C min^−1^ (for 5 min) with helium as a carrier gas at a flow rate of 1 mL·min^−1^ [22]. The injection port temperature was 250 °C. The sterols were analyzed as trimethylsilyl (TMS) ethers. The ergosterol and lanosterol were analyzed with reference to retention times and fragmentation spectra for standards. Other sterol TMS ethers were identified by comparison with the NIST database, or literature data, and quantitated using a standard curve for lanosterol.

### 4.8. The Plasma Membrane Fluidity Measurement (Laurdan Fluorescent Probe)

The PM fluidity was determined using a previously described method [22]. Laurdan is a fluorescent probe which is commonly used for determining the lipid packing in the PM [41]. Briefly, samples of the *C. albicans* WT, KS058, and 10C1B1I1 strains collected after 8 h and 24 h of culture were harvested and adjusted to A_600_ = 0.1 in 3 mL of PBS. The suspensions were incubated with a Laurdan fluorescent probe at a final concentration of 5 × 10^−6^ M for 20 min at RT in darkness. Laurdan probes were excited at λ = 366 nm (ex slit = 10 nm) and fluorescence spectra were recorded at 400–550 nm (em slit = 2.5 nm) using a fluorescence spectrophotometer equipped with a xenon lamp (HITACHI F-4500; manufacturer: Hitachi, Tokyo, Japan). The general polarization (GP) of the Laurdan probe incorporated in the PM of investigated strains was calculated using following formula:$$\mathrm{GP}=\frac{\sum_{425\;\mathrm{nm}}^{450\;\mathrm{nm}}\;IF-\sum_{475\;\mathrm{nm}}^{525\;\mathrm{nm}}\;IF}{\sum_{425\;\mathrm{nm}}^{450\;\mathrm{nm}}\;IF+\sum_{475\;\mathrm{nm}}^{525\;\mathrm{nm}}\;IF}$$
where: *IF* = fluorescence intensities determined for the Laurdan fluorescent probe at a certain range of wavelength.

### 4.9. Staining of Cell Wall and Structured Illumination Microscopy (SIM)

In order to determine the CW components, triple staining of *C. albicans* cells was performed. The mannans were stained using ConA dissolved in PBS containing 3% BSA, 1mM Ca^2+^, and Mn^2+^ and streptavidin conjugated with Alexa Fluor-568. Total chitin was visualized using CFW. The unmasked β-glucans were stained according to the protocol of Wagener et al. [42]. The *C. albicans* 24 h cultures were centrifuged and washed twice with PBS (4000× *g*, 5 min) and adjusted to A600 = 1 in PBS with 3% BSA. First, cells were incubated using 5 μg/mL of ConA for 1 h, 37 °C. The cells were washed twice with PBS and treated with 1:200 Alexa Fluor 568 conjugated streptavidin for 1 h, 37 °C. The cells were washed twice with PBS and 5 μg/mL Fc–hDectin-1 was added. After 1 h incubation, the cells were washed twice (4000× *g*, 5 min) and resuspended in PBS. The Fc-hDectin-1 treated cells were incubated with 1:250 Alexa Fluor 448-conjugated anti-human IgG Fc antibodies for 1 h, 4 °C. After that, the cells were washed twice as above, resuspended in PBS, and stained with CFW (0.025 mM) for 5 min, RT [43]. The cells were washed twice as above, resuspended in PBS, and concentrated. The observation of the preparations (at least 50 cells for each investigated strain) was performed using a super-resolution microscope (ZEISS Elyra 7 with Lattice SIM; Oberkochen, Germany).

### 4.10. Fluorescence-Activated Cell Sorting (FACS) Analyses

The *C. albicans* cell wall components were visualized as follows: the exposed β-glucan was stained with Fc–hDectin-1 (Section 4.9); the mannans were stained with 50 μg/mL ConA conjugated with FITC for 5 min, RT. The unmasked chitin was stained with WGA-FITC for 1 h, on ice [19]. After each staining, the cells were washed twice with PBS and in all cases, fixed with 3.7% formaldehyde for 15 min. After that, the cells were pelleted, washed twice with PBS, and resuspended in PBS. In order to perform the FACS analyses, the cell suspensions were diluted from 1:2 to 1:10, and 20,000 events were collected using a NovoCyte 2060R flow cytometer. Data were analyzed using NovoExpress software (ACEA Biosciences, San Diego, CA, USA). The fluorescent signal was obtained at a wavelength of 488 nm.

### 4.11. Statistical Analyses

For data analysis, statistical significance was determined using a Student’s t-test (binomial, unpaired). Data represent the means ± standard errors from at least three biological replicates.

## Figures and Tables

**Figure 1 ijms-23-01631-f001:**
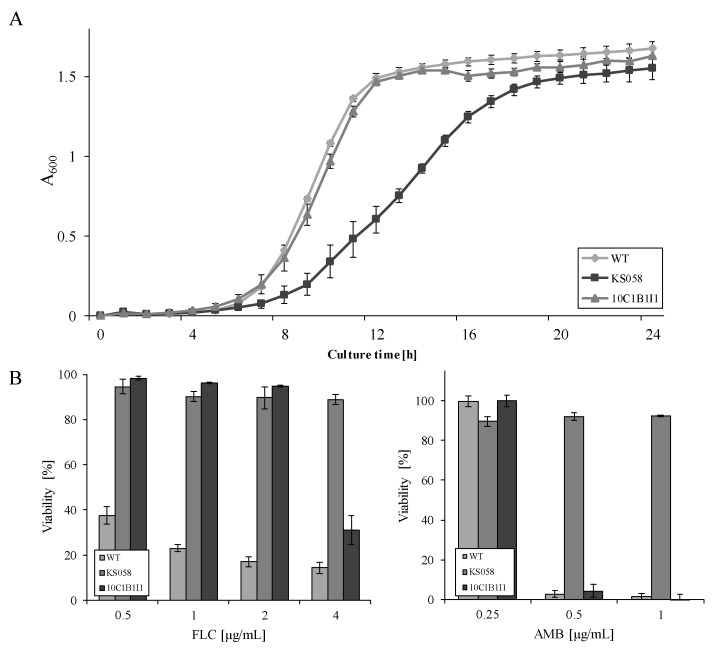
(**A**) The growth curve of the *C. albicans* SC5314 wild type (WT), *erg11Δ/Δ* (KS058) and the *ERG11^K143R/K143R^* (10C1B1I1) strains in a YPD medium (28 °C, 120 rpm; *n* = 3, ±SD). (**B**) The percent viability of the *C. albicans* WT, KS058, and 10C1B1I1 strains cultured for 24 h, 28 °C, in a YPD medium supplemented with FLC (left) or AMB (right) (*n* = 3, ±SD).

**Figure 2 ijms-23-01631-f002:**
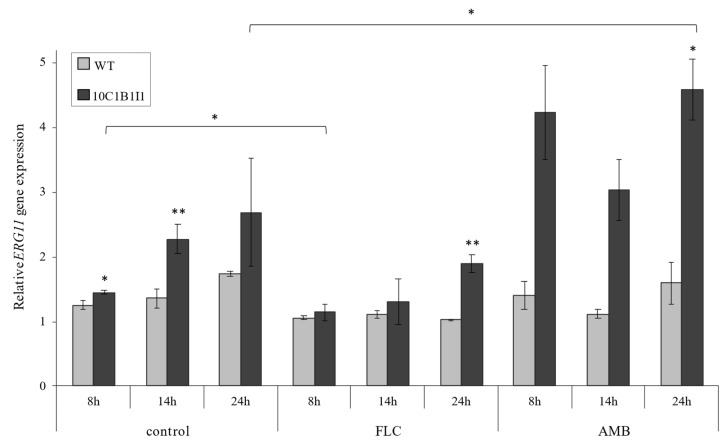
The relative *ERG11* gene expression in the *C. albicans* SC5314 wild type (WT) and *ERG11^K143R/K143R^* (10C1B1I1) strains determined for different growth times (8, 14, and 24 h) and culture conditions: control (YPD medium alone), and with the addition of FLC (4 μg/mL) or AMB (0.25 μg/mL); ±SD; *n* = 3. Statistical analysis was performed by comparing the expression of the *C. albicans* WT and 10C1B1I1, or between the different culture conditions (*, *p* < 0.05; **, *p* < 0.01).

**Figure 3 ijms-23-01631-f003:**
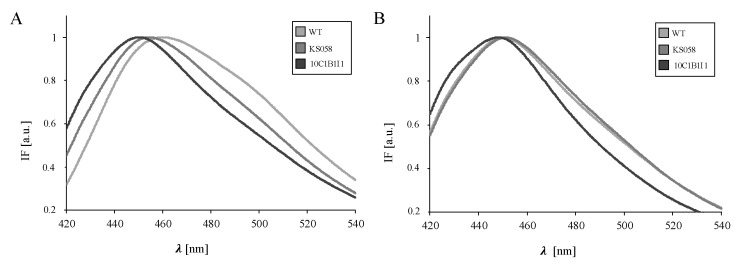
The representative, normalized fluorescence intensities (IFs) spectra for a Laurdan fluorescent probe incorporated into the plasma membrane of the *C. albicans* SC5314 (WT), *erg11Δ/Δ* (KS058), and *ERG11^K143R/K143R^* (10C1B1I1) strains in control conditions at 8 h (**A**) and 24 h (**B**).

**Figure 4 ijms-23-01631-f004:**
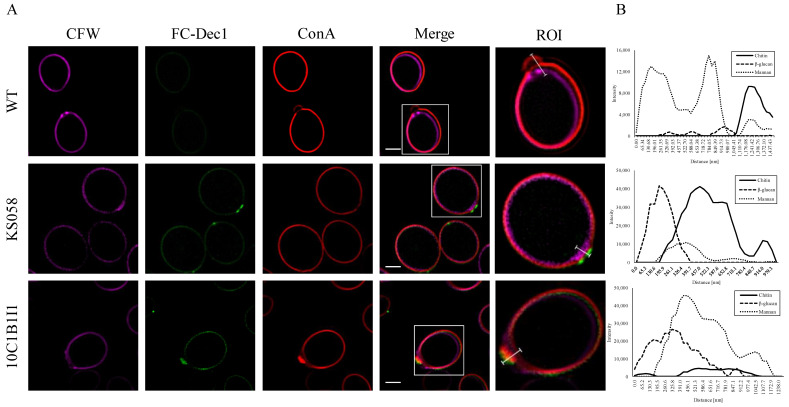
Triple staining of the *C. albicans* cell wall components for the SC5314 wild type (WT), *erg11Δ/Δ* (KS058), and *ERG11^K143R/K143R^* (10C1B1I1) strains (**A**). Total chitin was detected using calcofluor white (CFW) (purple). Exposed β-glucan was stained with Fc-hDectin-1 (FC-Dec1) and Alexa Fluor 448-conjugated anti-human IgG Fc antibodies (green). Mannans were detected using streptavidin and concanavalin A (ConA) conjugated with Alexa Fluor-568 (red). Scale bars of all presented images are equal to 2 μm. White squares on the merged picture represent region of interest (ROI). Fluorescence intensities for particular CW components for ROI (regions of bud scars; white lines) for investigated *C. albicans* strains (**B**). The distance [nm] between CW components is presented from outside to inside of the cell at the area of the bud scars.

**Figure 5 ijms-23-01631-f005:**
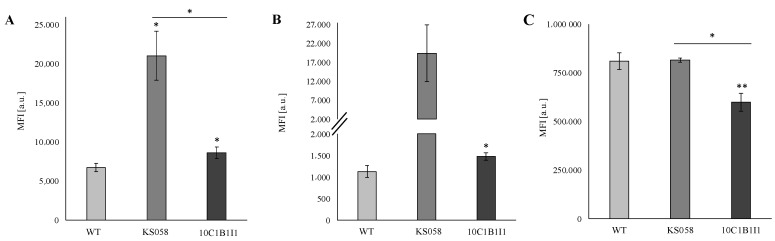
The exposure of chitin (panel (**A**)), β-glucan (panel (**B**)), and mannans (panel (**C**)) in the *C. albicans* SC5314 wild type (WT), *erg11Δ/Δ* (KS058) and *ERG11^K143R/K143R^* (10C1B1I1) strains grown for 24 h in YPD (28 °C, 120 rpm). In each case, the *C. albicans* cells were stained with wheat germ agglutinin conjugated with FITC (WGA-FITC; in case of exposed chitin, panel (**A**)), Fc-hDectin-1 and Alexa Fluor 448-conjugated anti-human IgG Fc antibodies (in the case of unmasked β-glucan, panel (**B**)) or streptavidin and concanavalin A (ConA) conjugated with Alexa Fluor 568 (in case of mannans, panel (**C**)) and analyzed by FACS. Median fluorescence intensities (MFIs) were quantified for three independent experiments. Statistical analyses were performed using the WT strain, or between KS058 and 10C1B1I1 (above lines) (*, *p* < 0.05 **, *p* < 0.01).

**Table 1 ijms-23-01631-t001:** The sterol analysis (μg/mg of the dry mass of isolated lipids; ±SD; *n* = 3; ND–not detected) determined for the *C. albicans* SC5314 wild type (WT), *ERG11^K143R/K143R^* (10C1B1I1), and *erg11Δ/Δ* (KS058) strains in different growth times (8, 14 and 24 h) and under different culture conditions: control (YPD medium alone), with the addition of FLC (4 μg/mL), or AMB (0.25 μg/mL). The sterol content was determined using the GC-MS method. Statistical analysis was performed by comparing the amounts of specific sterols in 10C1B1I1 or KS058 against the WT strain at each time point and under each culture condition (*, *p* < 0.05; **, *p* < 0.01; ***, *p* < 0.001).

	Ergosterol								
	Control			FLC			AMB		
	8 h	14 h	24 h	8 h	14 h	24 h	8 h	14 h	24 h
WT	6.95 ± 1.46	49.49 ± 13.12	28.82 ± 2.06	ND	ND	0.73 ± 1.27	31.78 ± 4.48	17.14 ± 0.84	20.20 ± 0.96
10C1B1I1	7.84 ± 3.73	93.82 ± 8.63 *	37.18 ± 1.53 **	5.83 ± 1.73 **	11.06 ± 0.71 ***	5.58 ± 3.10	39.82 ± 2.15 *	19.64 ± 2.73	16.19 ± 3.22
	Lanosterol								
	Control			FLC			AMB		
	8 h	14 h	24 h	8 h	14 h	24 h	8 h	14 h	24 h
WT	5.09 ± 2.11	2.52 ± 0.95	2.43 ± 0.36	8.73 ± 0.28	3.29 ± 0.77	5.87 ± 0.99	3.29 ± 0.08	4.92 ± 0.45	1.51 ± 0.10
10C1B1I1	5.11 ± 2.81	4.04 ± 0.93	2.31 ± 0.77	3.57 ± 0.99 ***	8.79 ± 2.33 *	5.72 ± 1.61	5.49 ± 1.60	1.77 ± 0.41 ***	1.45 ± 0.33
KS058	20.70 ± 0.43 **	11.45 ± 0.95 ***	22.53 ± 0.46 ***	26.06 ± 3.98 *	11.33 ± 1.45 **	53.34 ± 6.89 **	21.41 ± 3.25 ***	11.16 ± 2.44 *	20.37 ± 1.00 ***
	Eburicol								
	Control			FLC			AMB		
	8 h	14 h	24 h	8 h	14 h	24 h	8 h	14 h	24 h
WT	ND	ND	ND	2.88 ± 2.86	4.41 ± 3.91	3.41 ± 3.25	ND	ND	ND
10C1B1I1	ND	ND	ND	2.13 ± 2.43	1.97 ± 2.28	4.39 ± 4.43	0.17 ± 0.29	3.72 ± 6.32	ND
KS058	9.88 ± 0.95 **	8.12 ± 1.53 *	26.64 ± 2.14 **	9.41 ± 2.12 *	9.47 ± 1.11	40.07 ± 2.60 ***	10.32 ± 2.14 *	7.76 ± 0.88 **	19.48 ± 1.22 **
	14-α methylergosta-8-24(28)dienol								
	Control			FLC			AMB		
	8 h	14 h	24 h	8 h	14 h	24 h	8 h	14 h	24 h
WT	ND	ND	ND	ND	ND	3.41 ± 3.24	ND	ND	ND
10C1B1I1	ND	ND	ND	ND	ND	4.39 ± 4.43	ND	ND	ND
KS058	10.02 ± 0.32 ***	9.00 ± 0.82 **	13.31 ± 0.69 ***	17.06 ± 1.68 **	8.77 ± 1.64 *	26.12 ± 5.07 **	12.4 ± 1.81 **	15.12 ± 2.35 **	12.18 ± 1.52 **

**Table 2 ijms-23-01631-t002:** The general polarization values (GP; means ± SD, *n* = 6) for a Laurdan fluorescent probe incorporated into the plasma membrane of the *C. albicans* SC5314 (WT), *erg11Δ/Δ* (KS058), and *ERG11^K143R/K143R^* (10C1B1I1) strains. For the experiment, culturing was performed in a YPD medium (control conditions), at 28 °C, 120 rpm, for 8 h or 24 h, with the addition of FLC (4 μg/mL) or AMB (0.25 μg/mL). A statistical analysis was conducted in accordance to the GP values for the WT strain after specific culture times (*, *p* < 0.05; **, *p* < 0.01; ***, *p* < 0.001).

	Control		FLC		AMB	
	8 h	24 h	8 h	24 h	8 h	24 h
WT	−0.33 ± 0.046	−0.07 ± 0.111	−0.31 ± 0.049	−0.35 ± 0.026	−0.37 ± 0.038	−0.38 ± 0.029
KS058	−0.21 ± 0.026 ***	−0.09 ± 0.122	−0.29 ± 0.069	−0.34 ± 0.017	−0.37 ± 0.040	−0.36 ± 0.049
10C1B1I1	−0.10 ± 0.036 ***	0.05 ± 0.023 *	−0.26 ± 0.085	−0.32 ± 0.015	−0.28 ± 0.021 **	−0.35 ± 0.077

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
