# Peer review of "K143R Amino Acid Substitution in 14-α-Demethylase (Erg11p) Changes Plasma Membrane and Cell Wall Structure of Candida albicans"

_ijms, 2022, doi:10.3390/ijms23031631_

Round 1
Reviewer 1 Report
- In Fig1 X bar labels need to be corrected. It should be decimal instead of comma.
- The sentence (line 116-119) is confusing. Like the sentence mentioned that “required for a reduction in the viability”. It’s not clear how much reduction in growth- 20%, 50% or 100%? It is advisable that based on %growth or inhibition authors should mention it as IC50 or IC70 or IC80 or MIC.
- In Fig 2 legend - AmB conc. is too high i.e 0.25gm/ml. That’s several times higher than the MIC in C. albicans. It is surprising if cells survived up to 24h at this conc.
- It is interesting to see that ERG11 gene expression gradually increases from 8h to 24h in untreated 10C1B1I1 cells but total ergosterol level is highest at 14h (~12 fold higher than 8h) and further decreases at 24h (~2.5 fold in comparison to 14h). What would be the possible reason? Similarly, a significant decrease in ergosterol level was observed in both 14 h and 24 h of 10C1B1I1 growth after FLC treatment. What would be the possible reason for lower conc. of ergosterol after FLC and AmB treatment in ERG11 overexpressing 10C1B1I1 cells (Table1)?
- The authors showed that increased ergosterol contributes to chitin and beta-glucan exposure on the outside of the fungal cell in 10C1B1I1. Did authors observe echinocandins susceptibility in these cells because of exposed beta-glucans?
Author Response
Please, find our revised manuscript as well as our answers in the attachment.

Reviewer 2 Report
In this work, Derkacz and colleagues characterized a C. albicans strain bearing a point mutation in the gene erg11, coding a demethylase necessary for ergosterol biosynthesis. The point mutation is located in the codon in position 143, causing a lysine-arginine substitution (K143R). This mutation mimicks specific substitutions found in resistant strains of C. albicans. By comparing the phenotype of this point mutant with those of wild-type and erg11 deletion strains, the authors show that the K143R substitution in Erg11p causes resistance to fluconazol and amphotericin B, and correlate this observation with higher erg11 expression levels, higher ergosterol content in the plasma membrane, and redistribution of cell wall components glucans, chitin and mannans.
In my opinion, the manuscript is of interest for the readers of IJMS but there are a couple of points that need to be improved before acceptance. First, I recomment the authors to check and rewrite some sentences. I include a pdf copy of the manuscript with my comments. Some (there are more) of the points needing, in my opinion, rewriting are highlighted.
Second, some of the statements of the authors should be toned down. It is true that there is correlation among the point K143R mutation, resistance to FLC, erg11 expression levels, higher ergosterol concentration in the plasma membrane and remodeling of the cell-wall but, in my opinion and without more experimental data, it is just a correlation.
Finally, the results of Figure 4 and 5 are based on a couple of cells. Indeed, the cells analyzed in Figure 4 and in Figure 5 are the same (could be merged into just one figure). I understand that the results on the distribution of cell-wall components in these figures is supported by the FACS analysis in Figure 6 but the results in Figures 4/5 should be based on a significant number of cells. The authors should indicate how many cells have been analyzed through fluorescence microscopy and include quantitative data.
Taking everything into consideration my recommendation is major review.

Author Response
Please find our revised manuscript as well as our answers in the attachment.

Round 2
Reviewer 2 Report
The manuscript is now, in my opinion, ready for acceptance. I still recommend the authors to check some minor mistakes.